# A Strategy for Magnetic and Electric Stimulation to Enhance Proliferation and Differentiation of NPCs Seeded over PLA Electrospun Membranes

**DOI:** 10.3390/biomedicines10112736

**Published:** 2022-10-28

**Authors:** Irene Cuenca-Ortolá, Beatriz Martínez-Rojas, Victoria Moreno-Manzano, Marcos García Castelló, Manuel Monleón Pradas, Cristina Martínez-Ramos, Jorge Más Estellés

**Affiliations:** 1Center for Biomaterials and Tissue Engineering, Universitat Politècnica de València, Cno. de Vera s/n, 46022 Valencia, Spain; 2Neuronal and Tissue Regeneration Laboratory, Centro de Investigación Príncipe Felipe, 46012 Valencia, Spain; 3Biomedical Research Networking Center in Bioengineering Biomaterials and Nanomedicine, CIBER-BBN, 28029 Madrid, Spain; 4Unitat Predepartamental de Medicina, Universitat Jaume I, Avda/Sos Baynat, s/n, 12071 Castellón de la Plana, Spain

**Keywords:** polylactic acid, aligned substrates, neural progenitor cells, magnetic stimulation, electric stimulation

## Abstract

Neural progenitor cells (NPCs) have been shown to serve as an efficient therapeutic strategy in different cell therapy approaches, including spinal cord injury treatment. Despite the reported beneficial effects of NPC transplantation, the low survival and differentiation rates constrain important limitations. Herein, a new methodology has been developed to overcome both limitations by applying a combination of wireless electrical and magnetic stimulation to NPCs seeded on aligned poly(lactic acid) nanofibrous scaffolds for in vitro cell conditioning prior transplantation. Two stimulation patterns were tested and compared, continuous (long stimulus applied once a day) and intermittent (short stimulus applied three times a day). The results show that applied continuous stimulation promotes NPC proliferation and preferential differentiation into oligodendrocytic and neuronal lineages. A neural-like phenotypic induction was observed when compared to unstimulated NPCs. In contrast, intermittent stimulation patterns did not affect NPC proliferation and differentiation to oligodendrocytes or astrocytes morphology with a detrimental effect on neuronal differentiation. This study provides a new approach of using a combination of electric and magnetic stimulation to induce proliferation and further neuronal differentiation, which would improve therapy outcomes in disorders such as spinal cord injury.

## 1. Introduction

Spinal cord injury (SCI) is a serious clinical disease that can significantly reduce the quality of life of affected patients due to its associated partial or complete loss of sensory and motor function below the lesion site [1]. Regeneration of the adult central nervous system (CNS) is certainly limited, and no successful and effective treatment for SCI exists yet. The physical barrier created by a glial scar, a lack of neuron growth, and the presence of inhibitory molecules at the lesion site are the main factors beyond the absence of regeneration [2]. Several strategies have been addressed to manage and treat SCI, focusing on spine stabilization, the prevention of injury progression, and dealing with inflammation [3], but there is still little progress on improving SCI patient recovery [4].

Several cell types have been evaluated due to their capacity to treat SCI. Among them, there are Schwann cells, neural progenitor cells (NPCs), mesenchymal stem cells, and olfactory ensheathing cells [2]. NPCs used as a treatment for SCI and other CNS disorders have been broadly studied [3,5]. NPCs are multipotent cells residing in the CNS and can differentiate into neurons and glia (i.e., oligodendrocytes and astrocytes) [6]. There is evidence suggesting that NPC therapy shows potential in regenerating SCI damage, since cell transplantation and differentiation into glia and neurons assist in several aspects such as immunomodulation, neuroregeneration, and functional recovery through secreting growth factors and cytokines, regulating inflammation, inhibiting apoptosis of cells, and creating new synaptic connections, which in turn contribute to restoring neuronal networks [7,8]. It has been observed that NPC-derived glial cells confer several therapeutic benefits in supporting the regeneration, extension, and remyelination of injured axons and diminishing scar formation [9]. However, when applying NPC-based therapies, significant challenges still exist, such as poor cell engraftment and survival [7], as well as limited differentiation. In fact, the SCI lesion microenvironment causes NPCs to mostly differentiate into astrocytes, with minimal differentiation to neurons and oligodendrocytes, impending hence new neuronal circuits formation and remyelination [7,10,11]. Tissue engineering (TE) strategies have been gaining attention as an alternative approach in SCI injury treatment, with the aim of mimicking the native tissue both structurally and physiologically [12]. For this, TE usually relies on cell transplantation along with an engineered scaffold to improve their adhesion and survival, and may also include some type of cellular stimulation, which includes electrical [13], magnetic [14], mechanic [15], optogenetic [16], and chemical stimulation [17] and includes the use of growth factors, anti-inflammatory substances, and other molecules to influence cell behavior.

Scaffolds for SCI should meet a series of requirements to be suitable for implantation, including biocompatibility, biodegradability over time, mechanical properties that withstand spine and surrounding tissues forces, and an adequate architecture [6,18,19]. Synthetic biomaterials can be manufactured by controlled processes that lead to scaffolds with consistent properties adapted to specific applications, tailored design [20,21], and customized biodegradability and porosity [6]. PLA is a synthetic biomaterial widely used for medical and TE applications due to its good biocompatibility, ease of processing, slow degradation rate, and renal clearance of degradation products [22,23,24,25]. Several PLA-based devices have been approved by the Food and Drug Administration (FDA) due to its safety [25]. The scaffold itself can also intrinsically influence neural regeneration since there is evidence showing the micro-to-nanoscale topography importance not only in NPC adhesion, proliferation, and survival [26], but also in neuron growth promotion because of the geometrical cues provided [27]. It has been reported that aligned micro- and nano-fiber scaffolds can promote the regeneration of SCI, [13] guide the neurite outgrowth of dorsal root ganglia (DRG) [28], and increase the differentiation of embryonic stem cells and neural stem cells (NSCs) to neural lineages [29,30]. One of the most common techniques to produce this type of aligned scaffolds is electrospinning, which offers technical simplicity, tuneable properties, adaptability, and efficiency [31,32]. For these reasons, aligned PLA electrospun nanofiber membranes have been selected to seed NPCs in this study.

The capacity of electrical stimulation (ES) to promote the proliferation and differentiation of NPCs has been already demonstrated [33,34,35,36], as well as its capacity to either enhance neurite growth and axonal extension [31,37,38,39] or increase intracellular Ca^2+^ dynamics in vitro by means of the regulation of the cell membrane ionic channels [8,40]. In vivo, ES has been employed to promote neuroplasticity and functional regeneration after SCI [41,42]. ES presents advantages, such as a precise application time and voltage magnitude control, but also limitations, such as the need of electrodes to apply the electrical stimulus, which make it an invasive technique when used in vivo [13,43]. Alternatively, minimally invasive methods that are based on the external delivery of ES to transplanted cells have been developed, but other off-target effects may appear due to their low spatial resolution [11]. Thus, designing a strategy to apply ES directly to the damaged area is a challenge. Some studies have tried to develop implantable electronic devices, avoiding transcutaneous wires, but anatomical pockets need to be implanted [44,45]. The ES through the generation of a magnetic field would overcome these disadvantages, as recently shown by Han et al., since they developed a wireless strategy for applying ES based on electromagnetic induction [46].

Positive outcomes in clinics have been observed when using magnetic stimulation (MS) in different neurological disease treatments. MS have been used in the clinical treatment of depression, epilepsy, and insomnia and may be a future therapy of other pathologies, such as stroke, Alzheimer’s, Parkinson’s disease, and even SCI [1,47]. There is a growing interest in using this type of stimulus, with its non-invasive nature offering a substantial advantage in its application both in vitro and in vivo [1,48,49]. Several studies have shown the favourable effects that MS exerts on neural cells, although the mechanisms behind those positive MS outcomes are not completely understood. Beneficial outcomes of the clinical regeneration of nerves have been observed with low-frequency magnetic fields [50], and several studies show that magnetic field treatment promotes neurogenesis, neuronal differentiation, and neurite elongation and has protective and remodelling effects on cells and tissues [1,49,51,52,53]. In vitro, it has been observed that MS promotes either neurogenesis or NSC and NPC proliferation and differentiation into functional neurons [40]. It is thought that the mechanism underlying the response of cells to MS may be related to forces acting on macromolecules and charged particles in and around the cells [54], and some studies show Ca^2+^ ion changes induced by magnetic field exposure may be implicated in the observed effects [55,56].

This work presents an alternative and simpler approach tested in vitro, where an EMF generated by two Helmholtz coils induces an electric current in a golden loop, which will allow NPCs seeded in aligned PLA electrospun scaffolds to be electrically stimulated. Therefore, this approach will allow the NPCs to be stimulated not only electrically but also magnetically in a combined manner, avoiding direct contact of the sample and the wires. Two types of stimulation are studied: intermittent, by applying 2 h of stimulation three times per day for 3 days, and continuous, by applying 8 h of stimulation once a day for 3 days. This strategy for NPC conditioning could be useful in approaches including to improve efficacy of NPC transplantation for treatment of SCI.

## 2. Materials and Methods

### 2.1. Bioreactor Description

The bioreactor used to magnetically stimulate the device is made up of two coils facing each other along their axes, such that the magnetic field created by one coil reinforces that created by the other coil. If both coils are fed by a generator of alternating current (AC), an oscillating magnetic field will appear in the space between both coils. The winding of both coils must be done carefully so that the magnetic fields that they create reinforce each other and does not cancel them. The arrangement of the bioreactor can be seen in Figure 1A.

The magnetic field B created by one of these coils at a point P placed over the axis of the coil, at a distance x from its ending, outside of the coil, only depends on the intensity of the current flowing along the coil, I, the outer and inner radii of the coil, R_1_ and R_0_, its length L, and the number of windings, N, according to Equation (1).
(1)$$B=\frac{\mu_{0}\mathrm{NI}}{2L\left(R_{1}-R_{0}\right)}\left(\left(L+x\right)\ln\frac{R_{1}+\sqrt{R_{1}^{2}+{\left(L+x\right)}^{2}}}{R_{0}+\sqrt{R_{0}^{2}+{\left(L+x\right)}^{2}}}-x\ln\frac{R_{1}+\sqrt{R_{1}^{2}+x^{2}}}{R_{0}+\sqrt{R_{0}^{2}+x^{2}}}\right)$$

This equation can be simplified by using an average radius $R=\left(R_{0}+R_{1}\right)/2$. The magnetic field at point P can then be expressed as shown in Equation (2).
(2)$$B=\frac{\mu_{0}\mathrm{NI}}{2L}\left(\frac{L+x}{\sqrt{R^{2}+{\left(L+x\right)}^{2}}}-\frac{x}{\sqrt{R^{2}+x^{2}}}\right)$$

The magnetic field strongly depends on the distance *x* to the coil, decreasing when P moves away. For a more uniform magnetic field, a second coil is added. If d is the distance between the endings of both coils, the total magnetic field created by the couple of coils at a point P whose distance to a coil is x and d-x is its distance to the other coil, is given in Equation (3).
(3)$$\begin{array}{cc} B=B_{1}+B_{2}= & \frac{\mu_{0}\mathrm{NI}}{2L}(\left(\frac{L+x}{\sqrt{R^{2}+{\left(L+x\right)}^{2}}}-\frac{x}{\sqrt{R^{2}+x^{2}}}\right)\\ & +\left(\frac{L+d-x}{\sqrt{R^{2}+{\left(L+d-x\right)}^{2}}}-\frac{d-x}{\sqrt{R^{2}+{\left(d-x\right)}^{2}}}\right)) \end{array}$$

If the distance between both coils is of the order of the average radius of the coils, this magnetic field can be considered uniform in the space between them, near the axis of the coils. This magnetic field is directly acting on the culture cell, and it is one of the two stimuli applied to the culture. The second stimulus comes from an electric current induced thanks to Faraday’s law in a golden loop (low resistance and no cytotoxicity) placed inside the cell culture. This golden loop has a radius R_l_ perpendicularly to the axis of the coils. This loop is crossed by a magnetic flux created by the coils, and because of Faraday’s law, if this magnetic flux changes on time, an induced current flow along the golden ring also stimulates the cell culture. In our bioreactor, the current applied to the coils is an alternating current, and the magnetic field created by the coils will also be an alternating magnetic field. Therefore, the magnetic flux crossing the loop will be a variable magnetic flux. The amplitude of the current flowing along the loop, according to Faraday’s law, is shown in Equation (4), where B is the amplitude of the magnetic field created by the two coils, according to Equations (2) and (3); R_l_ is the radius of the loop, f is the frequency of the magnetic field (the frequency of the current applied to the coils), and R_g_ is the resistance of the loop.
(4)$$I=\frac{{B\pi R}_{l}^{2}2\pi f}{R_{g}}$$

Each coil has been built with a copper wire (diameter = 0.6 mm) coated with varnish and wound around a piece of a cylinder of polytetrafluoroethylene (PTFE) N = 1910 times. The inner radius of the coil is R_0_ = 1.75 ± 0.05 cm, the outer radius is R_1_ = 4.75 ± 0.05 cm, and its length is L = 3.0 ± 0.1 cm. The distance between the endings of both coils is d = 1.5 ± 0.1 cm, which is enough space to locate the Petri dish between both coils. The radius of the golden loop located inside the cell culture is R_l_ = 1.5 ± 0.1 cm (S = 7.1 ± 0.9 cm^2^), and its resistance is R_g_ = 0.5 Ω. Bioreactor coils were placed in a PTFE structure to guarantee either that the coils were always at the same distance or the correct culture dish colocation. A 35 mm culture dish was located between the two coils, perpendicularly to the coils’ axis, which in turn allowed the magnetic field created to stimulate the cells cultured inside the dish. The golden loop was achieved using gold wire of 99.99% purity (Electron Microscopy Science, 73,100), by twisting one wire end over the other.

### 2.2. PLA Membranes

Aligned PLA nanofiber membranes were obtained by electrospinning. PLA (10 wt %; Ingeo 40420 Resinex, Tarragona, Spain) was dissolved in dichloromethane/dimethylformamide in a 70/30 (*v*/*v*) proportion. The PLA solution was stirred until total PLA dilution at room temperature (RT). Briefly, electrospinning parameters were a voltage of 20 kV, a 20 cm distance between the needle and the collector center, a flow rate of 3 mL/h for 1.5 h, and a needle size of 30 G [57]. Obtained nanofibers had a diameter of around 600 nm. the membrane surface was morphologically characterized via a scanning electron microscope (FESEM; ULTRA 55, ZEISS Oxford Instruments, Wiesbaden, Germany), see Figure 1B. Platinum was employed to thinly recover the samples for its subsequent observation, and the voltage used was 1.5 kV. Final membranes were obtained by cutting these PLA membranes to a final size of 8 × 6 mm, being larger in the nanofiber’s direction.

### 2.3. Material Sterilization and Preconditioning

The bioreactor’s coils and structure as well as gold wire were sterilized by an autoclave at 121 °C for 30 min. PLA scaffolds were sterilized by means of ultraviolet (UV) radiation on both membrane surfaces for 1 h.

Before preconditioning, PLA membranes were washed 3 times in movement with ultrapure sterile water for 10 min. Scaffolds and golden loop preconditioning were achieved by immersion in high-glucose Dulbecco’s Modified Eagle Medium (DMEM) (L0102-500, Biowest, Nuaillé, France) supplemented with 1% penicillin/streptomycin (P/S) (15140122, Life Technologies, Carlsbad, CA, USA), following humid incubation at 37 °C and 5% CO_2_.

Prior to cell seeding, PLA scaffolds were coated with Matrigel^®^ (diluted at 1:20) (356234, Corning, Corning, NY, USA) for 1 h at 37 °C and 5% CO_2_. After that, membranes were rinsed twice with high-glucose DMEM supplemented with 1% P/S.

To assemble the 35 mm culture dish, firstly, the gold loop was placed inside, followed by the PLA membranes, which were equidistantly placed in the center of the culture dish, see Figure 1C. Therefore, PLA membranes were placed over, and thus touching, the golden loop. System assembly preceded cell seeding.

### 2.4. Cell Culture and Electric and Magnetic Stimulation

NPCs were obtained from the dissection of E-15 spinal cords from Sprague-Dawley rats in ice-cold Hank’s balanced saline solution (HBSS) supplemented with P/S, and the tissue was mechanically dissociated by repetitive pipetting. Isolated NPCs were expanded as neurospheres in a growth medium in Ultra Low Attachment plates (3471, Corning), and Passages 5–9 were used for differentiation experiments. The growth medium consisted in a NeuroCult™ proliferation medium (05700, Stemcell technologies, Grenoble, France) supplemented with NeuroCult™ Proliferation Supplement (05701, Stemcell Technologies), 1% P/S, 0.7 UI/mL heparin (H3393, Sigma-Aldrich, St. Quentin Fallavier Cedex, France), 20 ng/mL epidermal growth factor (EGF) (PHG0311, Thermo Fisher, Horsham, UK), and 20 ng/mL basic fibroblast growth factor (bFGF) (13256-029, Thermo Fisher). Neurospheres were dissociated before seeding using Accutase (L0950-100, Biowest) following the manufacturer’s instructions. Dissociated NPCs were seeded in 2 drops (6 × 10^5^ cells/3 µL drop) to promote their dispersion around the PLA membrane. NPCs were cultured over a total of 7 days in a differentiation culture medium, changing it on Day 3 of culture. The first differentiation medium used was composed of DMEM/F-12 (Sigma-Aldrich, D6421-500ML) supplemented with 1% P/S, 2 mM L-glutamine, 5 mM HEPES buffer, 0.125% NaHCO_3_, 0.6% glucose, 0.025 mg/mL insulin, 80 μg/mL apotransferrin, 16 nM progesterone, 60 μM putrescine, 24 nM sodium selenite, 4% *w*/*v* BSA, 4% heparin, and 20 ng/mL bFGF. The second differentiation medium was composed of DMEM/F-12 (D6421-500ML, Sigma-Aldrich) supplemented with 1% P/S, 2 mM L-glutamine, 5 mM HEPES buffer, 0.125% NaHCO_3_, 0.6% glucose, 0.025 mg/mL insulin, 80 μg/mL apotransferrin, 16 nM progesterone, 60 μM putrescine, 24 nM sodium selenite, and 2% FBS. 

For MS and ES, a power supply provided a sinusoidal alternating wave of 7.86 V peak to peak (V_pp_) that was injected into the set of coils (V_rms_ = 2.78 V), which results in a current of 47 ± 0.1 mA flowing along each coil at a frequency of 75 Hz. According to Equation (3), the amplitude of the theoretical magnetic field in the space between both coils should be 1.24 mT. This calculation was verified by measuring the magnetic field with a Hall sensor, resulting in an amplitude of 1.13 ± 0.01 mT for the experimental magnetic field. The existent difference between the calculated and measured value of the magnetic field may be due to the fact that the created magnetic field is not perfectly uniform, in addition to other experimental measurement errors. However, the calculated and measured values are close enough to allow the assumption that Equation (4) can be used to estimate the induced current. Therefore, using the measured value of the magnetic field would induce an electric current of 750 µA of amplitude along the loop. Depending on the duration and pattern of the application, two different stimulation types, continuous and intermittent, were tested to determine its influence on NPCs (Figure 1D). Continuous stimulation consisted in stimulating the cells 8 h/day and leaving 16 h of repose, whereas intermittent stimulation was applied in cycles of 2 h of stimulation and 6 h of repose, 3 times a day. The time of application was 3 days in both patterns, and they both started one day before NPC seeding and ended on Day 4 of the experiment. A control group had the same conditions but without stimulation. The cell culture was repeated three times to observe three independent experiments results.

### 2.5. Immunocytochemistry

After 7 days on a differentiation culture, samples were prepared for an immunostaining assay. Prior to fixation, seeded membranes were rinsed with 0.1 M phosphate buffer (PB) (D9564, Sigma-Aldrich), and cells were fixed using 4% PFA for 15 min, followed by 2 washes of 5 min with 0.1 M PB. Permeabilization and blocking steps were done with 10% normal goat serum (NGS) (50062Z, Thermo Fisher) and 0.1% Triton X-100 (T8787, Sigma-Aldrich) in 0.1 M PB for 1.5 h at RT. Samples were incubated overnight at 4 °C with primary antibodies against *beta-III-tubulin* (1:400; 11-264-C100, Exbio, Vestec, Czech Republic), *Oligodendrocyte transcription factor 2* (*Olig2*) (1:400; AB9610, Sigma-Aldrich), *Glial Fibrillary Acidic Protein* (*GFAP*) (1:1000; PA1-10004, Thermo Fisher), *Nestin* (1:200; ab6142, Abcam, Cambridge, UK) and *Ki67* (1:400; GTX16667, Gene Tex, Irvine, CA, USA). On the next day, respective conjugated secondary antibodies against primary antibody species were used: goat anti-mouse 488, goat anti-rabbit 488, goat anti-chicken 555, and goat anti-rabbit 647, (1:200; Thermo Fisher). Incubation occurred for 2 h at RT in darkness. Cell nuclei were stained by incubation with 4′,6-diamidino-2-phenylindole (DAPI) in 0.1 M PB (1:1000; D9564, Sigma-Aldrich) for 10 min. 

Images were acquired using a confocal microscope (Zeiss LSM780 Confocal Microscope, Oberkochen, Germany), and consistent exposures were used. For image quantification, ImageJ software (version 2.1.0, NIH, Bethesda, MD, USA) was employed to quantify the number and percentages of differentiated cells, and the NeuronJ plugin from ImageJ (version 1.4.3) was used to measure neurite lengths. All neurites coming out of the neurons’ soma were measured, and among them, the longer path was selected. Neurons whose neurites came out of the immunostaining image were not considered.

### 2.6. Statistical Analysis

Results are expressed as mean ± standard error of the mean (SEM). GraphPad Prism software was used for the analysis of the data. A Shapiro–Wilk test was used to test the normality of the data distribution. If normality was met, a nested and ordinary one-way ANOVA test was performed to compare between groups (Tukey’s multiple comparisons correction). If data were not normally distributed, the Kruskal–Wallis non-parametric test was used (Dunn’s multiple comparisons correction). Differentiation data have been obtained from three independent experiments and at least 3 samples of each experiment have been analysed. Significant differences between groups are indicated by *, **, ***, or **** when the *p*-value is below 0.05, 0.01, 0.001, and 0.0001, respectively.

## 3. Results and Discussion

### 3.1. Bioreactor and System Features

The system setup is shown in Figure 1A. The bioreactor is explained in more detail in the Materials and Methods section. Briefly, the bioreactor design consisted of two coils in which an alternating current (75 Hz, 7.86 V peak to peak) was injected to generate a magnetic field of 1.13 mT. For ES, a golden loop was placed in a 35 mm culture dish, which was situated between both coils (Figure 1B). The NPCs were seeded onto aligned PLA electrospun nanofiber membranes disposed onto the golden loop, as shown in Figure 1C. Thus, the generated magnetic field induced an alternating current of 75 Hz and 750 µA at the golden loop. The culture dish disposition can be seen in Figure 1C. Therefore, all of the NPCs of the same culture dish share the same culture media, as well as secreted factors. NPCs were cultured in differentiation conditions for 7 days, and a combined magnetic and electric stimulation was applied for 3 days, testing two different stimulation patterns, continuous and intermittent (Figure 1D).

In the first stages of the experiments, a different material loop with higher resistance was located inside the cell culture. Therefore, under the same created magnetic field, the induced current in the loop was much lower. When induced current values close to those used in this study were looked for with that loop, the current injected into the coils needed to be much higher, leading to the heating of the coils, and reaching temperatures of 40 °C in the culture dish (measurements made with the set placed inside an oven at 37 °C). This heating clearly would affect the cell culture by killing the cells. The solution to this problem was found in using a golden loop instead. Some tests using this golden loop were carried out in order to measure the heating of the coils at 37 °C. Employing a generated magnetic field of 3.7 mT (around three times the used magnetic field in this study), no heating higher than 0.1 °C was detected for 60 min in the culture dish zone, showing that, under these conditions, the temperatures of the cell culture will not be affected. In addition to this, typical electrical stimulation frequencies range from 1 to 200 Hz, and some studies found better results within the low-medium frequency spectrum values [58,59,60]. Due to its low resistance, gold as a material selection for the loop makes it possible to generate elevated induced currents using not only low currents injected into the coils (thus avoiding their heating) but also frequencies at the low-medium spectrum range.

Because of the disposition of the membranes into the same culture dish, all of the cells in different membranes share the same culture media, which means that some molecules liberated by the cells of one membrane might affect the other cells seeded on the resting membranes. However, this does not affect the conclusions obtained in this study, since all of the study groups were under the same experimental conditions, so it can be concluded that the effects observed between groups are due to the magnetic and electric stimulation exposition. The fact that electrical current was flowing through the golden loop was confirmed, since in previous experiments, under the same experimental conditions, a golden loop was placed over the seeded membranes, making more direct contact with cells, and results showed a clear reduction in cell viability. This viability reduction could only be due to the generated electric current in view of the fact that the employed magnetic field value was not changed (1.13 mT). Although the induced electric current value could be reduced by altering the parameters of the sinusoidal current injected into the coils, the same parameters were needed to compare it to the previous experimental results. For this reason and with the aim of reducing this detrimental effect of the electric current, the golden loop was placed under PLA-seeded membranes.

In this study, an electrical stimulus was applied employing neither electrodes nor an electroconductive material. No electroconductive material was incorporated into the non-conductive PLA scaffold because, despite the normal conditions, conductive materials, such as polypyrrole or graphene, are highly conductive; under physiological or alkaline pH, their conductivity is highly reduced [61]. Thus, culture media conductivity is higher than the conductivity of the usually employed conductive materials in those conditions. Therefore, since the entire system is immersed in culture media, the induced current would not flow by the conductive scaffold, but by the culture media. The use of a non-conductive material not only does not impede the electrical stimulation of the NPCs seeded on it, but also allows one to avoid using those conductive materials in the scaffold fabrication, as toxic effects of conductive polymer degradation products and negative long-time biocompatibility have been observed [62,63]. Due to PLA membrane non-conductivity, two possible mechanisms by which electricity stimulates the seeded NPCs have been hypothesized. Firstly, as previously mentioned, it is possible that some currents flow through the culture media. Secondly, the cells seeded closer to the golden loop may be more affected by the induced current than the more distant ones, so paracrine signaling may occur from closer cells into neighboring cells. However, more research would be needed in order to validate these hypotheses.

Recently, Han et al. [46] used a similar strategy of electrically stimulating NSCs by electromagnetic induction. They used only one coil, as well as a conductive annular graphene scaffold where cells were seeded in order to directly apply electrical stimulation to them. Although they needed high frequencies, of the order of 20 kHz, to have enough current to electrically stimulate cells due to the high resistance of graphene, they obtained positive results with this stimulation system, demonstrating that wireless ES does not affect animal survival and that the system effectively promotes NSC differentiation to neurons.

As a future perspective and with the aim of using this strategy in vivo, some scaffold modifications will be required. A scaffold may be improved in various manners, such as including some magnetic nanoparticles or incorporating the golden loop directly to the scaffold.

### 3.2. Continuous Stimulation Promotes NPC Proliferation and Differentiation into Neurons and Oligodendrocytes Progenitors

After 7 days of differentiation culture, cell density, proliferation and differentiation were studied by immunostaining assays and quantification analysis. Cell density was evaluated as the number of cell nuclei per mm^2^ in immunocytochemistry images. For cell proliferation, the ratio (in percentage) of *Nestin*+ and *Ki67*+ double positive cells to *Nestin*+ cells were calculated (*Nestin*+ and *Ki67*+/*Nestin*+). Differentiation was studied calculating the percentage of neurons (*β-III-tubulin*+/DAPI), oligodendrocytes progenitors (*Olig2*+/DAPI), and astrocytes (*GFAP*+/DAPI).

Figure 2 shows immunocytochemistry images for *Nestin*+ and *Ki67*+ cells and quantification analysis of cell density and embryonic proliferating cells. Under observation at the microscope, a higher cell density was noted at continuous stimulation group. This higher cell density was corroborated later by analysing the number of cell nuclei per mm^2^, Figure 2D. This higher cell density was corroborated by analysing the number of cell nuclei per mm^2^ (Figure 2D). It was found that cell density in the continuous stimulation group (7702 ± 243 nuclei/mm^2^) was significantly higher than the control (5632 ± 122 nuclei/mm^2^) group and the intermittent stimulation (5226 ± 134 nuclei/mm^2^) group. In fact, continuous stimulation cell density was 37 and 57% higher than the control and intermittent stimulation cell density, respectively. Intermittent stimulation did not present significant differences when compared to the control. Proliferation analysis was performed via *Nestin* and *Ki67* markers after 7 days on differentiation culture (Figure 2E). *Nestin* is a marker of cellular stemness, while *Ki67* is a marker of active cell proliferation. These two markers were used to evaluate the percentage of embryonic cells that were proliferating after 7 days of differentiation culture. This parameter was evaluated by means of the ratio between both *Nestin*+ and *Ki67*+ double positive to *Nestin*+ cells (in percentage). Even under conditions of induced differentiation, continuous stimulation group (15.1 ± 1.1%) presented a significantly higher percentage of embryonic proliferating cells when compared to control (6.8 ± 1.4%) and intermittent stimulation group (5 ± 0.6%). Embryonic proliferating cells percentage in continuous stimulation is two and three times higher than those cells in control group and intermittent stimulation group, respectively. Although these proliferation percentages are relatively low, immunostaining assay was performed after 7 days in differentiation culture. Despite this, continuous combined magnetic and electric stimulation may affect somehow the proliferation of the embryonic NPCs. The fact that intermittent stimulation does not affect embryonic cells proliferation may indicate that both stimulation and repose times of this intermittent pattern are not long enough to allow cells to proliferate and to secrete all necessary molecules. More research about this fact would be necessary in order to elucidate the concrete underlying reasons of these observed results.

Representative images of oligodendrocyte progenitor-, neuron-, and astrocyte-differentiated cells after 7 days in the differentiation culture in all three study groups are shown in Figure 3A–C. The percentage of cells differentiated to oligodendrocyte progenitors and neurons was found significantly increased in the continuous stimulation group. The percentage of *Olig2*-positive cells is shown in Figure 3D. Continuous stimulation (25.5 ± 0.7%) prompted a significantly higher *Olig2*+ percentage in comparison to the control cells (17.9 ± 0.6%) and intermittent stimulation cells (19.8 ± 0.8%). No significant differences were found between the intermittent stimulation and control groups. Figure 3E shows the percentage of cells differentiated to *β-III-tubulin* positive neurons. The continuous stimulation percentage (2 ± 0.1%) showed a significant increase of 65% when compared to the control group (1.2 ± 0.1%) and was also more than three times higher than the intermittent stimulation group percentage (0.6 ± 0.1%), indicating that a continuous pattern is helpful in promoting neuronal differentiation. It was also noted that intermittent stimulation significantly reduces the differentiation to neurons when compared to the unstimulated group, which denote an unfavorable effect of an intermittent stimulation pattern. There were no significant differences in the astrocyte percentage between the continuous (15.1 ± 0.8%), intermittent (14.6 ± 0.6%), and control (13.2 ± 0.5%) groups (Figure 3F).

Electric and magnetic fields can influence different aspects of NPC development, including cell proliferation and differentiation. On the one hand, ES has been shown to promote NPC proliferation [64,65], neuronal differentiation, and process elongation [33,66,67]. Chang et al. [34] stimulated NSCs using a biphasic electrical current and noted an increase in cell proliferation and differentiation into *NeuN*-, *MAP2*-, and *β-III-tubulin*-positive neurons. On the other hand, several studies have found that MS increases the ratio of differentiated cells to neurons and promotes neurite outgrowth [55,68,69], in addition to improving NPC and NSC proliferation [51,70,71]. Ma et al. studied two patterns of magnetic stimulation in NSCs, intermittent (5 min on–10 min off) [72] and continuous (4 h) [56]. Although they found the mRNA levels of proneural genes (*Math1*, *Math3*, *Neurogenin1*, and *Tuj1*) to be upregulated in the intermittent stimulation, this exposure type did not significantly change cell proliferation or neuronal differentiation, results that were similar to those found by Nikolova et al. [73] in NPC magnetic stimulation (5 min on–30 min off). Instead, continuous stimulation not only significantly increased NSC proliferation (elevated expression of *Sox2*, *Hes1*, and *Hes5* genes), but also promoted neuronal differentiation and neurite outgrowth. Although Ma et al. [56] only employed a magnetic stimulation to stimulate NSCs, their results support those of this study in terms of stimulation pattern times, so that cells may prefer continuous stimuli and long repose times rather than shorter intermittent stimuli and repose times. Although the intermittent stimulation times used here were longer and more continuously applied than the ones applied by Ma et al. and Nikolova et al., our intermittent stimulation seems to be insufficient to achieve positive results in NPC proliferation and differentiation, and it may be detrimental for neuronal differentiation. Moreover, only continuous stimulation group has an increased percentage of embryonic proliferating cells after 7 days of differentiation culture. As to intermittent stimulation pattern, two hours of stimulation might not be enough to positively stimulate cells, and 6 h of repose between stimulations might not be enough time for cells to secrete all necessary molecules and to produce the changes needed to effectively increase proliferation and differentiation.

The lesion microenvironment after transplantation provokes the differentiation of NPCs predominantly to astrocytes, with reduced differentiation to neurons or oligodendrocytes [10]. The results presented in this study may improve this situation, since an increase in neuronal and oligodendrocytic differentiation was observed with a continuous stimulation pattern, a situation that would encourage neuronal network formation and their remyelination, which is important for restoring the conductance of axons and preventing their degeneration.

### 3.3. Continuous Magnetic and Electric Stimulation Affects Astrocyte Maturation into Different Morphological Subtypes

Astrocytes are usually divided into two subtypes based on morphologic and molecular criteria, protoplasmic and fibrous [74]. Morphological differences were noted in astrocytes after continuous stimulation exposition, as shown in Figure 4. Based on the morphological description of the astrocyte’s subtypes and simply by observing the samples in a microscope, a clearly different morphology between groups was noted. The control (Figure 4A) and intermittent (Figure 4B) stimulation groups had mainly protoplasmic-like astrocytes (Figure 4D,E; signalled with an arrow), which are characterized by many densely-packed branches that likely touch many synapses, playing an important role in neuromodulation [75]. Instead, the continuous stimulation group (Figure 4C) had predominantly fibrous-like astrocytes (Figure 4F; signalled with an arrow), which are distinguished because their ramifications are lesser, longer, and straighter than protoplasmic ones. Fibrous astrocytes are present along with white matter tracts, contacting Ranvier nodes and hence contributing to homeostasis [75]. The proportion of each astrocyte subtype was quantified based on these morphological differences between them, calculating the ratio of polygonal and fibrous astrocytes over the total amount of *GFAP*-positive cells (Figure 4G,H). Morphologically, protoplasmic astrocyte ratio was significantly higher in the control (75.6 ± 1.5%) and intermittent (78.8 ± 1.3%) groups than in the continuous one (36 ± 0.9%), while the fibrous astrocyte ratio was significantly superior in the continuous stimulation (64 ± 0.8%) group than in the control (24.5 ± 1.5%) and intermittent (21.2 ± 1.3%) groups. Based on these results, we can hypothesize that continuous stimulation may induce a fibrous-like morphology. However, this is a qualitative analysis and a more extended astrocytic study using phenotypic markers will be needed in order to confirm this hypothesis. Some studies have found this morphological difference after stimulus application. For example, Yang et al. [76] found that human astrocytes displayed a more elongated morphology after direct current (DC) exposure in a dose-dependent manner. Giraldo et al. [7] also found fibrous morphological changes in the astrocyte population after stimulating optogenetically *ChR2*-NPCs, while protoplasmic astrocytes were more present in the unstimulated groups.

### 3.4. Continuous Magnetic and Electric Stimulation Promotes Neurite Growth

To evaluate whether the simultaneous magnetic and electric stimulation during differentiation of NPCs somehow affects axon development in neuron cells, the neurite length of each neuron was determined using the NeuronJ plugin (Figure 5). Microscope images of the *β-III-tubulin* marker showed that continuous stimulation leads to a largely interconnected network of neurons in comparison to the unstimulated and intermittent stimulation groups (Figure 5A–C). Furthermore, the mean and maximum axon length per neuron in all three study groups was calculated (Figure 5D,E). The results of neurite length quantification showed not only that continuous stimulation axons were longer on average (69.6 ± 3.2 µm) than control axons (49 ± 3 µm), but also that the lengthiest continuous stimulated axons (98.1 ± 4.6 µm) had extended more than those in the control group (66.9 ± 5.3 µm). Continuous stimulation did not show significant differences when compared to the intermittent stimulation mean axon length (59.7 ± 9.4 µm) and intermittent maximum axon length (103.4 ± 17.4 µm). No significant differences between the intermittent stimulation and control groups were found. In terms of mean axon length per neuron, continuously stimulated neurites extended 17% more so than the intermittent ones and 42% more so than the control ones. Therefore, although intermittent magnetic and electric stimulation seems to impair neuronal differentiation, continuous stimulation not only increases the percentage of *β-III-tubulin*-positive neurons, but their axons had also extended more.

Many investigations have shown the positive effects of electrical stimulation and magnetic stimulation on neurite outgrowth. For instance, Fu et al. [77] used pulsed electric currents to stimulate NSCs and found that it promoted neurite elongation and neuron differentiation, while Ma et al. [56] found that the magnetic stimulation of NSCs also increased the ratio of differentiation to neurons, as well as promoted axonal extension. Several studies have been conducted to elucidate the underlying mechanisms of the magnetic and electric stimulation effects in NPCs, but they are still not completely understood [49,78]. These effects have been hypothesized to be related to several factors, such as an increase in intracellular Ca^2+^ (since it has an important role in cell fate determination), an overexpression of voltage gate calcium channels (VGCCs) and transient receptor potential canonical 1 (TRCP1), an activation of several cellular pathways, an augment in the expression of BDNF, an upregulation of proneural genes, among others [8,36,40,52,56,79,80,81]. Therefore, although the complete picture of the underlying mechanisms remains to be fully clarified, what is clear is that ES and MS influence cellular behaviour via several mechanisms, affecting hence functional activities of the cells (including proliferation and differentiation). Thus, different stimulation patterns will lead to different cell fate, so that stimulation parameters determination is crucial in order to obtain the desired results. This can be problematic since there are many parameters to consider, such as stimulation duration, times of application, field magnitude, frequency, type of stimulation, among others. In this study, we propose two stimulation patterns using a nonconventional strategy that allows to combine two common stimulation types, electric and magnetic. Intermittent stimulation pattern has not shown differences in cell density, proliferation or differentiation of NPCs, being detrimental to neuronal differentiation. Authors hypothesize that the cellular mechanisms activated by these stimulation pattern do not finally affect NPCs proliferation, but negatively affect neuronal differentiation pathways. However, continuous stimulation shows good performance in NPCs proliferation and differentiation. To elucidate the concrete mechanisms that activate combined magnetic and electric stimulation, additional research is needed.

Both electric and magnetic stimulation have been applied as a non-invasive technique to modulate the excitability of the brain [82,83,84]. ES locally applied at the injured zone of the spinal cord has been widely studied, and some preclinical studies provide promising results regarding the promotion of functional recovery and regeneration [11,13,85,86]. The main inconvenience related to electrical stimulation arises in in vivo application, since electrodes are directly inserted into the stimulation site. This invasiveness is a source of infection and can lead to a loss in stimulation efficacy and changes in the pH, among others [11,44]. Therefore, the strategy of inducing an electric current using a magnetic field allows one to apply not only electrical stimulation directly to the lesion site without using invasive electrodes but also a simultaneous application of two different types of stimulation widely used individually in clinics and broadly studied in research [8,40,52,78,82,83]. In this study, the combined application of electrical and magnetic stimulus to NPCs seeded in a PLA membrane was achieved, without employing neither electroconductive material nor magnetic particles, which are commonly used in the application of both stimuli, respectively [14,31,40,54,77]. Encouraging results of this combined stimuli were observed here in vitro. Continuous electric and magnetic stimulation showed positive effects when promoting NPC proliferation and differentiation into oligodendrocytes progenitors and neurons, also enhancing neurite elongation, outcomes that could improve neuronal circuit formation and remyelination in SCI regeneration. Moreover, based on the qualitative analysis of the astrocytic morphology, we hypothesize that continuous stimulation may affect astrocytes maturation to a fibrous-like morphology subtype. As a future perspective, more research will be needed to introduce elements that allow one to improve the scaffold and the astrocytes phenotypic maturation in sight of the potential this strategy has in SCI in vivo treatment and regeneration.

Taking together all the results presented here, in addition to those obtained by Han et al. [46], there is evidence supporting the effectiveness of the method of electric current stimulation by magnetic field induction in neural stem and progenitor cell therapy. The use of this strategy added to an engineered scaffold that supports cells growth and survival can lead to a reduction of the invasiveness of current SCI treatment approaches, being thus a possible future therapeutic strategy for this pathology.

## 4. Conclusions

This study presents a suitable strategy of in vitro NPC stimulation using a combination of both magnetic and electrical stimuli (without wires and electrodes), by inducing an electric current in a golden loop. Intermittent stimulation does not provoke changes in proliferation and reduces differentiation to neurons. In contrast, continuous stimulation enhanced NPC proliferation and differentiation to oligodendrocytes progenitors and neurons. Although more research is needed to further evaluate the in vitro and in vivo effects of the combination of both stimulation types to NPCs, this study offers evidence of the validity of this approach, providing a possible new strategy for cell therapy in SCI and other neurological disease treatment.

## Figures and Tables

**Figure 1 biomedicines-10-02736-f001:**
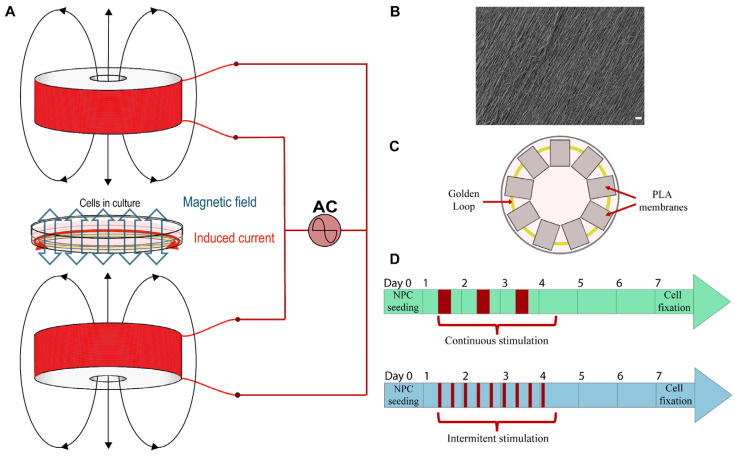
(**A**) Bioreactor arrangement scheme: an alternant sinusoidal current is injected into two coils to create an oscillating magnetic field that is directly applied to a culture dish, inside which a golden loop is placed, resulting in the generation of an induced current in this loop. (**B**) A representative scanning electron microscope image of an aligned PLA electrospun scaffold. Scale bar: 20 µm. (**C**) Scheme of the 35 mm culture dish assembly: the sterile golden loop is firstly placed inside the dish, and, after that, the PLA membranes, where the NPCs will be seeded, are positioned equidistantly from the culture dish center and over the golden loop. (**D**) Scheme showing the combined magnetic and electric stimulation protocol employed for continuous and intermittent stimulation pattern (in red is shown the stimulus application). Stimulation parameters in both continuous and intermittent stimulation were a frequency of 75 Hz, a magnetic field of 1.13 mT, and an induced current in the loop of 750 µA, applied for 3 days.

**Figure 2 biomedicines-10-02736-f002:**
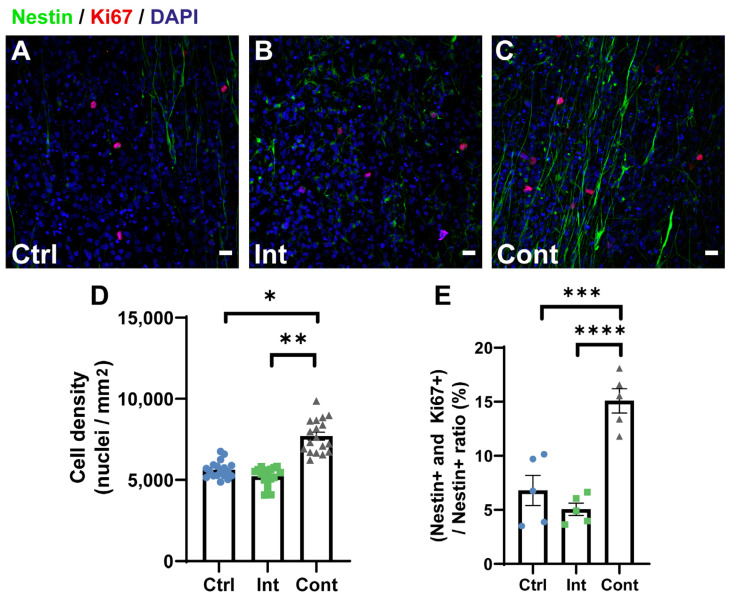
(**A**–**C**): representative images of immunocytochemistry staining for *Nestin* (green), *Ki67* (red) and DAPI (blue) of the control (**A**), intermittent stimulation (**B**), and continuous stimulation (**C**) groups. Scale bars: 20 µm. (**D**,**E**): Quantification results of cell density and embryonic proliferating cells percentage (*Nestin*+ and *Ki67*+/*Nestin*+) for the control (Ctrl), intermittent stimulation (Int), and continuous stimulation (Cont) groups after 7 days of differentiation culture. (**D**) Cell density quantification results (Ctrl: 5632 ± 122 nuclei/mm^2^; Int: 5226 ± 134 nuclei/mm^2^; Cont: 7702 ± 243 nuclei/mm^2^). Statistical differences tested by a nested one-way ANOVA. (**E**) Percentage of embryonic proliferating cells (Ctrl: 6.8 ± 1.4%; Int: 5 ± 0.6%; Cont: 15.1 ± 1.1%). Statistical differences tested by ordinary one-way ANOVA. Data are expressed as mean ± SEM. * *p* < 0.05; ** *p* < 0.01; *** *p* < 0.001; **** *p* < 0.0001.

**Figure 3 biomedicines-10-02736-f003:**
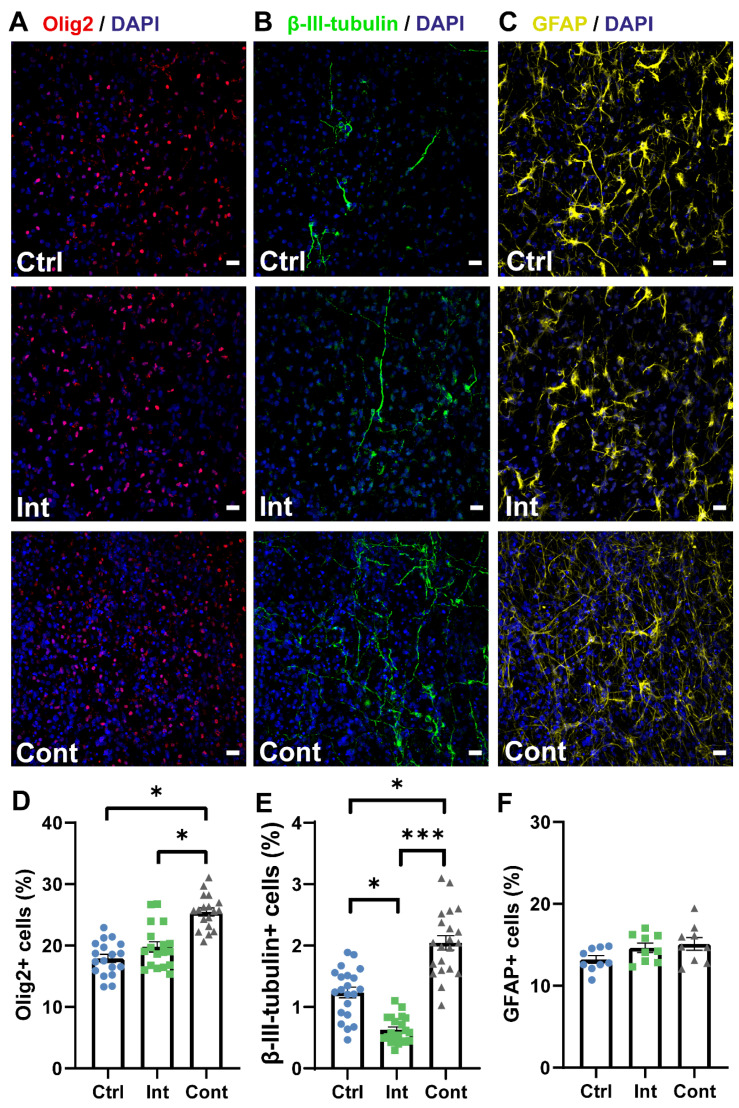
(**A**–**C**): Representative images of immunocytochemistry staining for DAPI (blue) and *Olig2* (red) (**A**), *β-III-tubulin* (green) (**B**), and *GFAP* (yellow) (**C**) of the control, intermittent stimulation, and continuous stimulation groups. Scale bars: 20 µm. (**D**–**F**): Quantification results of the oligodendrocyte percentage (*Olig2*-positive cells/DAPI), neuron percentage (*β-III-tubulin*-positive cells/DAPI), and astrocyte percentage (*GFAP*-positive cells/DAPI), respectively, for the control (Ctrl), intermittent stimulation (Int), and continuous stimulation (Cont) groups. (**D**) Percentage of *Olig2*+ cells (Ctrl: 17.92 ± 0.65%; Int: 19.79 ± 0.83%; Cont: 25.47 ± 0.66%). (**E**) Percentage of *β-III-tubulin*+ cells (Ctrl: 1.24 ± 0.09%; Int: 0.63 ± 0.05%; Cont: 2.04 ± 0.12%). (**F**) Percentage of *GFAP*+ cells (Ctrl: 13.21 ± 0.47%; Int: 14.63 ± 0.57%; Cont: 15.11 ± 0.78%). Statistical differences tested by a nested one-way ANOVA. Data are expressed as mean ± SEM. * *p* < 0.05; *** *p* < 0.001.

**Figure 4 biomedicines-10-02736-f004:**
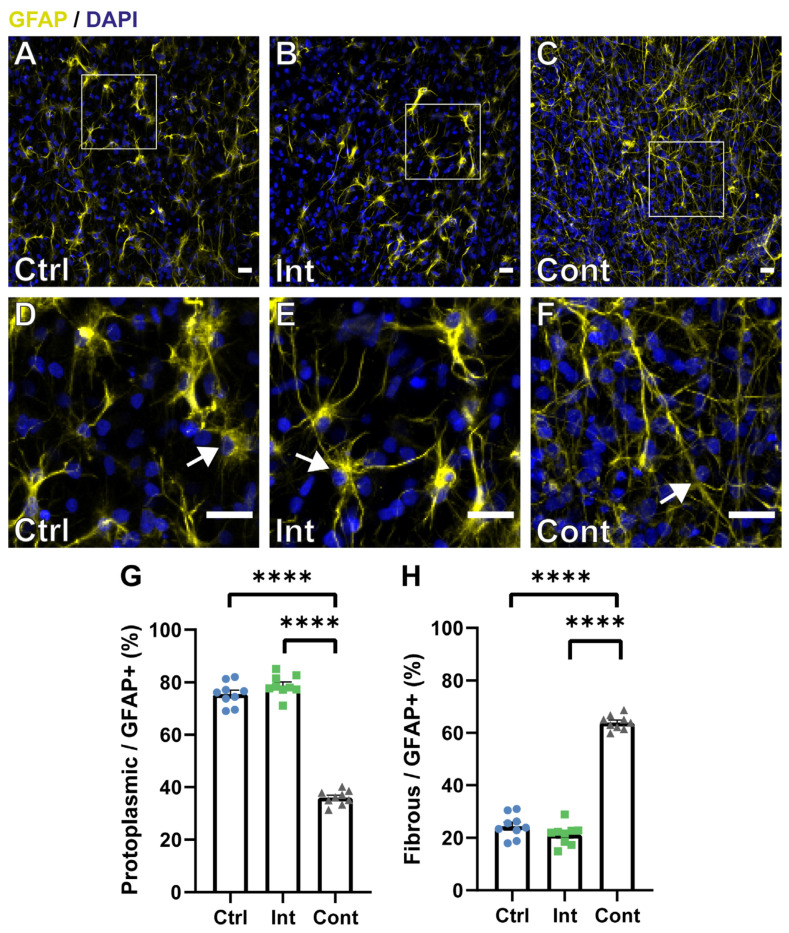
(**A**–**C**): Representative images of immunocytochemistry staining for *GFAP* (yellow) and DAPI (blue) of the control (**A**), intermittent stimulation (**B**), and continuous stimulation (**C**) groups. White squares indicate the shown magnifications (**D**–**F**). (**D**–**F**): Magnifications of the control (**D**), intermittent stimulation (**E**), and continuous stimulation (**F**) groups. White arrows indicate the group characteristic astrocyte morphology, showing protoplasmic (**D**,**E**) and fibrous (**F**) astrocytes. Scale bars: 20 µm. (**G**,**H**): Quantification analysis of the protoplasmic (**G**) and fibrous (**H**) astrocyte ratio in the control (Ctrl), intermittent stimulation (Int), and continuous stimulation (Cont) groups. (**G**) Protoplasmic astrocyte ratio quantification (protoplasmic astrocytes/*GFAP*-positive cells) (Ctrl: 75.6 ± 1.5%; Int: 78.8 ± 1.34%; Cont: 36 ± 0.9%). (**H**) Protoplasmic astrocyte ratio quantification (fibrous astrocytes/*GFAP*-positive cells) (Ctrl: 24.5 ± 1.5%; Int: 21.2 ± 1.3%; Cont: 64 ± 0.8%). Data are expressed as mean ± SEM. **** *p* < 0.0001. Statistical differences tested by a nested one-way ANOVA.

**Figure 5 biomedicines-10-02736-f005:**
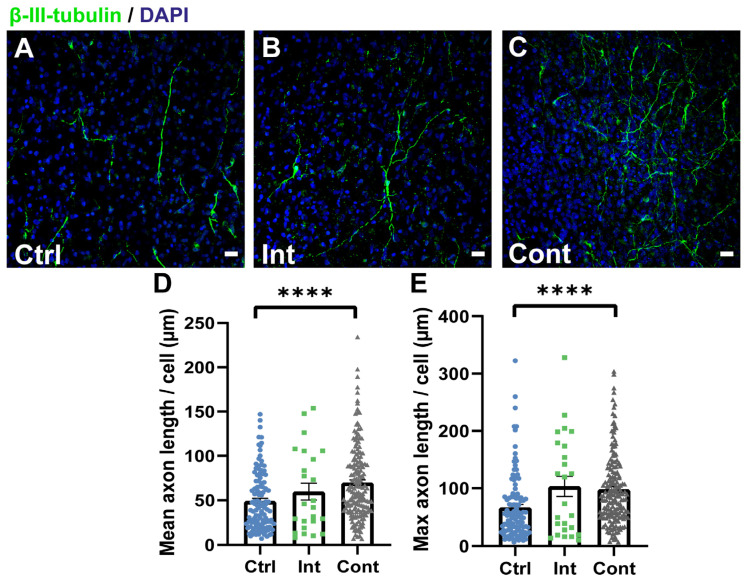
(**A**–**C**): Representative images of immunocytochemistry staining for *β-III-tubulin* (green) and DAPI (blue) of the control (**A**), intermittent stimulation (**B**), and continuous stimulation (**C**) groups. Visual inspection of the images shows that continuous stimulation neurons had extended more than the other groups. Scale bars: 20 µm. (**D**,**E**): Quantification results of the mean axon length per neuron and the maximum axon length per neuron in µm, respectively, for the control (Ctrl), intermittent stimulation (Int), and continuous stimulation (Cont) groups (number of neurons quantified: 115 Ctrl; 25 Int; 175 Cont). (**D**) Mean axon length per neuron in µm (Ctrl: 49.08 ± 3.06 µm; Int: 59.69 ± 9.4 µm; Cont: 69.61 ± 3.17 µm). (**E**) Maximum axon length per neuron in µm (Ctrl: 66.86 ± 5.31 µm; Int: 103.4 ± 17.4 µm; Cont: 98.12 ± 4.62 µm). Data are expressed as mean ± SEM. **** *p* < 0.0001. Statistical differences tested by Kruskal–Wallis (**D**,**E**). Quantification analysis of neuronal extension confirmed what was observed in the immunostaining images.

## Data Availability

Not applicable.

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
