# Peer review of "A Strategy for Magnetic and Electric Stimulation to Enhance Proliferation and Differentiation of NPCs Seeded over PLA Electrospun Membranes"

_biomedicines, 2022, doi:10.3390/biomedicines10112736_

Round 1
Reviewer 1 Report
This work by Cuenca-Ortolá and colleagues face a topic that has already been described in the literature. The use polymer is different and as well as the bioreactor. In my opinion the work can be better classified as method paper more than research paper, since any functional readout is given. I underline here some observations:
1- in general in the figures should be added the name of antibody as authors did in figure 3.
2-ki67 in figure 2 is completely a-specific. I cannot see the real antibody presence inside the nuclei of the cells. If these are the best pictures, the result is not good.
3-it is not clear the difference between the expression of beta III tub highlighted in figure 3 and figure 5, specifically in the cont+ condition
4- could you please explain better what you mean in the discussion that this model is useful for cell therapy? How will be this possibile? Do you mean to take the differentiated cells and implant them? for which pathology? do you think the cells will be stay alive and proliferate? This is not clear to me
Reviewer 2 Report
This study provides a new approach of using a combination of electric and magnetic stimulation to induce proliferation and further neuronal differentiation, which would improve therapy outcomes in disorders such as spinal cord injury This study is a descriptive study, and it would be more convincing and better if the experiment involved the mechanism. The author should provide the specific percentage value of increase of proliferation and neuronal differentiation
Reviewer 3 Report
In the present study, authors investigated whether magnetic and electric stimulation affected the phenotypes of neural stem/progenitor cells (NSPC). Although idea is interesting, the mechanism by which magnetic and electric stimulation influenced the traits of these cells remains unclear. In addition, the following issues should be revised.
1. Fig2: Authors claimed that intermittent stimulation did not affect NSPC proliferation. However, they showed that the percentage of Ki67+ cells was significantly increased compared with controls (Fig. 2E). How do authors mean by that? Authors should clarify this point and revise the text accordingly. And also, the traits of Ki67+ cells remain unclear. To assess the changes of NSPC proliferation after stimulation, I recommend that authors evaluate the “ratio of nestin+ and Ki67+ double positive cells/total nestin+ cells” and/or “ration of Sox2+ and Ki67+ double positive cells/total Sox2+ cells”.
2. Fig2E: “I-St” and “C-St” should be changed into “Int” and “Cont” as shown in other Figures (e.g., Fig2D, Fig3D-F, Fig4G,H).
3. Fig3A,D: As for the evaluation of oligodendrocytic differentiation, authors used Olig2. However, Olig2 is generally known as a marker of OPC (oligodendrocyte progenitor cells) rather than mature oligodendrocytes. I recommend that authors use other oligodendrocyte markers, such MBP and MAG.
4. Fig3B, E: Authors claimed that intermittent stimulation decreased the ratio of β-III-tubullin+ cells compared with controls. In contrast, continuous stimulation increased the ratio of β-III-tubullin+ cells compared with controls. Why was generation of new neurons decreased after intermittent stimulation? And also, authors only used β-III-tubullin+ cells as a marker for neurogenesis. I recommend that authors use other neuronal markers, such as MAP2 and synaptophysin.
5. Fig4: Authors claimed that intermittent stimulation did not affect the ratio of protoplasmic or fibrous GFAP+ astrocytes compared with controls. In contrast, continuous stimulation decreased the ratio of protoplasmic GFAP+ astrocytes, while it increased the ratio of fibrous GFAP+ astrocytes. However, meaning of this word remains unclear. And also, it seems that it is difficult to count protoplasmic or fibrous astrocytes exactly by morphological shapes without specific markers for them.
Round 2
Reviewer 3 Report
I think authors significantly improved manuscript.